# An investigation of a hundred COVID-19 cases and close contacts in Ethiopia, May to June, 2020: A prospective case-ascertained study

Shambel Habebe Watare[1]*, Mikias Alayu Alemu[1], Adamu Tayachew[1], Negussie Yohannes[1], Lehageru Gizachew[2], Adisu Kebede[1], Asdesach Tessema[1], Wubayehu Kassa[1], Mengistu Biru[1], Mikias Teferi[2], Naod Wendrad[2], Kirubel Eshetu[1], Gizaw Teka[1], Habtamu Yimer[2], Faiqa Ebrahim[3], Dagnachew Mulugeta[3], Alemnesh Mirkuze[1], Saro Abdela[1], Musa Emmanuel[4], Abdulhafiz Hassen[1], Mesfin Wosen[1], Mukemil Hussen[1], Yaregal Fufa[1], Getachew Tolera[1], Zewdu Assefa[1], Aschalew Abayneh[1], Ebba Abate[1]

1 Ethiopian Public Health Institute, Addis Ababa, Ethiopia, 2 Ministry of Health, Addis Ababa, Ethiopia, 3 World Health Organization Country Office, Addis Ababa, Ethiopia, 4 World Health Organization Regional Office for Africa, Brazzaville, Congo

* shambelhw@gmail.com

**Data Availability Statement:** Relevant data are within the paper and its Supporting Information files.

## Abstract

### Background

Corona Virus Disease 2019 is a novel respiratory disease commonly transmitted through respiratory droplets. The disease has currently expanded all over the world with differing epidemiologic trajectories. This investigation was conducted to determine the basic clinical and epidemiological characteristics of the disease in Ethiopia.

### Methods

A prospective case-ascertained study of laboratory-confirmed COVID-19 cases and their close contacts were conducted. The study included 100 COVID-19 laboratory-confirmed cases reported from May 15, 2020 to June 15, 2020 and 300 close contacts. Epidemiological and clinical information were collected using the WHO standard data collection tool developed first-few cases and contacts investigation. Nasopharyngeal and Oropharyngeal samples were collected by using polystyrene tipped swab and transported to the laboratory by viral transport media maintaining an optimal temperature. Clinical and epidemiological parameters were calculated in terms of ratios, proportions, and rates with 95% CI.

### Result

A total of 400 participants were investigated, 100 confirmed COVID-19 cases and 300 close contacts of the cases. The symptomatic proportion of cases was 23% (23) (95% CI: 15.2%-32.5%), the proportion of cases required hospitalization were 8% (8) (95%CI: 3.5%-15.2%) and 2% (95%CI: 0.24% - 7.04%) required mechanical ventilation. The secondary infection rate, secondary clinical attack rate, median incubation period and median serial interval were 42% (126) (95% CI: 36.4%-47.8%), 11.7% (35) (95% CI: 8.3%-15.9%), 7 days (IQR:

**Funding:** This study was conducted as part of the routine national COVID-19 response efforts under the EPHI. The costs, materials and supplies used for this study was covered by the EPHI as part of the pandemic response efforts. There was no other institution that had role in study desing, data collection and analysis, decision to publish, or preparation of the manuscript. Thus, the author(s) received no specific funding for this work. The funders had no role in study design, data collection and analysis, decision to publish, or preparation of the manuscript.

**Competing interests:** All authors declared that they have no competing interests.

4–13.8) and 11 days (IQR: 8–11.8) respectively. The basic reproduction number ($R_O$) was 1.26 (95% CI: 1.0–1.5).

## Conclusion

The proportion of asymptomatic infection, as well as secondary infection rate among close contacts, are higher compared to other studies. The long serial interval and low basic reproduction number might contribute to the observed slow progression of the pandemic, which gives a wide window of opportunities and time to control the spread. Testing, prevention, and control measures should be intensified.

## Background

Corona Virus Disease 2019 (COVID-19) is a severe acute respiratory infection (SARI) that emerged in December 2019 in Wuhan, China. The outbreak was declared as a Public Health Emergency of International Concern (PHEIC) by the World Health Organization (WHO) on 30 January 2020 [1]. The disease rapidly spread all over the world in a short period and causes millions of cases and hundreds of thousands of deaths [2]. Yet, still, the true magnitude of the pandemic is underestimated mainly because a substantial proportion of people with the infection are undetected either because they are asymptomatic or have only mild symptoms, and hence, they do not visit health facilities and limited testing capacity. The case fatality rate of the disease varies across the countries from less than 0.1% to over 25% [3].

The causative agent for COVID-19 is Severe Acute Respiratory Syndrome Corona Virus 2 (SARS-CoV-2), which is a positive-sense, single-stranded RNA virus belonging to the Beta coronavirus B lineage and is closely related to the SARS-CoV virus [4].

SARS CoV-2 is transmitted from infected individuals to healthy individuals mainly through respiratory droplets via sneezing, coughing, and talking without covering the mouth and the nose. Expelled droplets during these acts may linger in the air and infect individuals that come into contact with them in an enclosed space and during aerosol-generating procedures in the health care setting. The virus can be transmitted from a pre-symptomatic or asymptomatic infected individual. Infected individuals can transmit the disease up to two weeks of first symptom onset [5–8].

Studies from China and other countries including the USA and Europe indicated that the clinical spectrum of COVID-19 appears to be wide, comprising mild type without pneumonia, common type with pneumonia, severe type with respiratory distress, and critical type with respiratory failure, shock, or even death [9]. In China, up to 43% of patients required oxygen, and up to 6% required mechanical ventilation. Fever and cough were the most common clinical sign and symptoms of COVID-19 cases [10, 11]. Perhaps, a significant proportion of patients with confirmed SARS CoV-2 infection remains asymptomatic, which are cases potentially missed by detection strategies and put a threat to infection control via person to person contact [12].

However, as with many novel respiratory pathogens, key epidemiological, clinical, and virologic parameters of COVID-19 are not fully known. Therefore, this study is aimed at determining epidemiological and clinical parameters of COVID-19 in Ethiopia, as part of the WHO solidarity study [13]. This will facilitate timely estimates of the severity and transmissibility of COVID-19 infection. The finding will also inform public health responses and policy decisions in developing appropriate response strategies.

## Methods

### Design

This is a few cases and close contacts investigation which was a prospective case-ascertained study of laboratory confirmed COVID-19 cases and their identified close contacts.

### Study population and period

The study includes 100 COVID-19 laboratory confirmed cases reported from May 15, 2020 to June 15, 2020 from Addis Ababa City, the first epi-center of COVID-19, and the capital of Ethiopia. Among the close contacts of the 100 cases, three of each respective case contacts were randomly selected using a lottery method and followed-up. Contacts were included in the study regardless of the contact settings, household, health facility, workplace and other contacts. Hence, the study includes 100 cases and 300 close contacts which made total study participants of 400.

Cases and close contacts were identified based on the standard WHO confirmed COVID-19 case and close contact definitions. Each enrolled case and close contact were monitored for 14 days starting from the date of detection and last date of exposure to COVID-19 confirmed case respectively.

### Data collection

Data were collected using Open Data Kit (ODK). The customized version of the standard first few cases and close contacts (FFX) data collection tool from the World Health Organization was uploaded to ODK and data collection was done through face-to-face interviews of cases, close contacts, and their caretakers or families. Also, review of patient medical records was done to check for comorbidities and other medical conditions.

For the cases, clinical and epidemiological information was collected on the date of positive laboratory diagnosis for SARS CoV-2 and the 14[th] day of detection. For close contacts, epidemiological and clinical information was collected on the date of identification and the 14[th] day since their last date of exposure to their respective index case. Besides, data from the symptom diary of each case was compiled during the 14 days follow-up period to check if they develop any sign and symptoms of COVID 19. If the close contacts develop any sign and symptom of COVID 19 before the 14th day of follow up, PCR test will be done and treated accordingly.

Data collectors were Medical Doctors and Epidemiologists trained on the data collection tool and the data collection application. The data were synchronized to the Ethiopian Public Health Institute (EPHI) server every day. The quality, consistency, and completeness of data were checked by a data management specialist at Ethiopian Public Health Institute (EPHI).

### Laboratory investigation

Nasopharyngeal and oropharyngeal samples were collected from cases and close contacts on the first and last days of follow-up (i.e. on the 14[th] day). Samples were collected by polystyrene tipped and plastic shaft swab and transported by a 3 milliliter viral transport media (VTM) at a temperature of 2˚C to 8˚C on daily basis. RT-PCR test of the nasopharyngeal and oropharyngeal samples was done immediately following arrival to the National Influenza Laboratory at the EPHI and results were issued within 24 hours of receipt.

### Statistical analysis

The data were extracted from the EPHI server in Microsoft excel format and extensive cleaning was done. After it was cleaned, exported to Statistical Package for Social Science (SPSS)

version 23 for analysis. Continuous variables were expressed as medians with interquartile range (IQR) and simple ranges. Categorical variables were analyzed in ratios, proportions, and rates. Two-sided 95% CIs was also determined for rates and proportions of epidemiological and clinical indicators.

Secondary infection rate was calculated by dividing the number of contacts tested positive for COVID 19 by total contacts included in the study. Secondary clinical attack rate was calculated by dividing the number of contacts develop COVID 19 sign and symptoms and tested positive for COVID 19 by total contacts included in the study. Incubation period for each case was determined by counting the days from last date of contact with COVID 19 case to the first date of onset of COVID 19 sign and symptoms.

Serial interval was estimated as the number of days between the data of onset of the index case and the date of onset of its close contact who develop sign and symptoms for COVID 19. Basic reproduction number (Ro) is calculated by dividing the number of contacts tested positive for COVID-19 by total (100) cases included in the study.

### Patient and public involvement

This study is not a clinical trial. It was conducted following expedited public health act as part of pandemic response operations to generate scientific evidence to inform the response efforts. Thus, patients or the public were not involved in the design, or conduct, or reporting, or dissemination plans of our research.

### Ethical approval

The study is part of the global multi-center solidarity study and used the solidarity study protocol which was approved by the ethical review board of the World Health Organization Headquarters.As a result, the ethical approval process was waived and conducted as part of COVID 19 pandemic response efforts under the Ethiopian Public Health Institute.

The interviewers had explained the objectives, the process, and the benefits of the study for each study participant. Each participant was asked for their informed consent to participate in the study. A written consent was obtained from all study participants. In the case of interviewing children, the consent was obtained from their parents or guardians. All study participatns were considered for the study based on their informal consents. Interview using a structured data collection tool was conducted for those who consented to take part in the study.

Personal identifying information was not used during analysis and report writing. The records were kept confidential and will not be shared with a third party.

## Result

### Socio-demographic characteristics

A total of 400 participants, 100 confirmed COVID-19 cases, and 300 close contacts were investigated. Sixty-one (61%) of the confirmed cases were males. Eighty-nine (89%) of the confirmed cases and 214 (75.5%) of the close contacts were in the age range of 15 to 49 years.

Health care workers constitute 6% (6) of cases and 2.3% (7) of close contacts. About three percent of the health care worker contacts were tested positive by RT-PCR for COVID-19 (Table 1).

### Clinical and epidemiological characteristics of cases and close contacts

In this study, the symptomatic proportion of the cases was 23% (23) (95% CI: 15.2%-32.5%), the proportion of cases that required hospitalization among the symptomatic cases was 8%

**Table 1. Socio-demographic characteristics of the cases and close contacts, Ethiopia, 15 May to 15 June 2020.**

| Variable | | Cases (%) | Contacts | | Total participants (%) |
|---|---|---|---|---|---|
| | | | Total (%) | Positive (%) | |
| **Gender** | **Male** | 61 (61%) | 150 (50%) | 63 (42%) | 211(52.8%) |
| | **Female** | 39 (39%) | 150 (50%) | 63 (42%) | 189 (47.2%) |
| **Age** | **<15** | 2 (2%) | 49 (16.3%) | 20 (40.8%) | 51 (12.7%) |
| | **15–49** | 89 (89%) | 214 (71.3%) | 93 (43.5%) | 303 (75.8%) |
| | **50–70** | 9 (9%) | 32 (10.7%) | 11 (34.4%) | 41 (10.2%) |
| | **>70** | 0 (0%) | 5 (1.7%) | 2 (40.0%) | 5 (1.3%) |
| **Occupation** | **Health worker** | 6 (6%) | 7 (2.3%) | 2 (2.9%) | 13 (3.3%) |
| | **Non health worker** | 94 (94%) | 293 (97.7%) | 124 (97.1%) | 387 (96.7%) |

(95%CI: 3.5%-15.2%) and 2% (95% CI: 0.24% - 7.04%) of the hospitalized cases were required mechanical ventilation.

Among 23 symptomatic cases, 12 (52.2%) had cough, eight (34.8%) had sore throat, seven (30.4%) had fever and three (13%) had shortness of breath.

Eight percent (95% CI: 3.5%-15.2%) of the COVID-19 cases had underlying morbidities. Diabetes Mellitus (DM) was the commonest underlying medical condition (n = 3, 3%) followed by cardiovascular diseases (n = 2, 2%). Obesity, HIV AIDS, chronic liver disease and chronic kidney disease are also accounts for 1% each among COVID 19 patients.

On the 14th day following admission, 72% (72) (95%CI: 3.5%-15.2%) of the COVID-19 cases had tested negative and recovered from the disease. However, 1% (1) (95%CI: 0.03%-5.45%) was died and 27% (27) (95%CI: 18.61%-36.80%) of them were not recovered at 14th day of admission.

From the 300 close contacts, 42% (126) (95% CI: 36.4%-47.8%) were tested positive by RT-PCR for SARS-COV-2 infection. However, only 11.7% (35) (95% CI: 8.3%-15.9%) had at least one clinical sign and/or symptoms of COVID-19. The median incubation period for the symptomatic contacts who turned out to be positive for COVID-19 was seven days (IQR: 4–13.8).

The serial interval was calculated for ten clusters of symptomatic cases and close contacts. The ten clusters were selected for SI calculation because there were contacts tested positive in those ten clusters and transmission of disease from index to contacts is required to perform SI estimation/calculation. In these clusters, there were 10 symptomatic primary cases and 23 symptomatic secondary cases from close contacts. Based on these figures, the median serial interval was calculated to 11 days (IQR: 8–11.8) (Table 2).

The highest secondary infection rate was reported among household contacts (45.3%, 95% CI: 38.6%-52.1%) and the lowest was reported among workplace contacts (30.9%, 95% CI: 17.6%-47.1%) (Table 3).

## Discussion

This case ascertained study of a few cases and close contacts provides valuable clinical and epidemiological characteristics of COVID-19 cases and close contacts in Ethiopia. The study revealed that the asymptomatic proportion of SARS COV-2 is significantly higher and the commonest symptoms were cough and sore throat followed by fever and shortness of breath. The majority of COVID-19 cases recovered by the end of the second week of admission.

The large asymptomatic proportion (77%) in our study is inconsistent with a study conducted among 104 patients in Diamond Princess Cruise Ship, which reported the asymptomatic proportion of cases to be 41% on admission and 32% at the end of their follow-up period

**Table 2. Clinical and epidemiological indicators for the cases and close contacts investigation in Ethiopia, 15 May to 15 June 2020.**

| Indicators | Results |
|---|---|
| Hospitalization rate | 8% (95%CI: 3.5%-15.2%) |
| Mechanical ventilation required | 2% (95%CI: 0.24% - 7.04%) |
| Case-hospitalization ratio [CHR] | 11.5:1 |
| Symptomatic proportion of cases | 23% (95% CI: 15.2%-32.5%) |
| Asymptomatic fraction of infection | 77% (95% CI: 67.5%-84.8%) |
| Secondary clinical attack rate of COVID-19 infection among close contacts | 11.7% (95% CI: 8.3%-15.9%) |
| Secondary infection rate of COVID-19 infection among close contacts | 42% (95% CI: 36.4%-47.8%) |
| The incubation period in days | Median = 7 (IQR: 4–13.8), Range (2–14) |
| Serial interval in days | Median = 11 (IQR: 8–11.8), Range (5–26) |
| Case-fatality ratio [CFR]. | 1% (95% CI: 0.03%-5.45%) |
| Basic reproduction number ($R_0$) | 1.26 (95% CI: 1.0–1.5) |
| Reproduction ratio (R) | 1:1.26 |
| Herd Immunity threshold | 20.6% |

[10]. Another study in China also indicated that only 2.1% of peoples with SARS-CoV-2 infection were symptom-free and it was 33.3% in South Korea [14, 15]. According to a systematic review and meta-analysis of 41 studies, only 15.6% of COVID-19 cases were asymptomatic [12].

Cough is the commonest symptom in the present study where 52% of the symptomatic COVID-19 patients have manifested. Chih-Cheng Lai and colleagues who summarized the clinical characteristics of COVID-19 cases in China and outside, find out fever as the commonest symptom followed by cough [16]. Another literature on COVID-19 has also shown that 80% to 90% of the cases were presented with fever, while cough was manifested by 50% of the cases [6].

The present study revealed that the case fatality rate is 1% in Ethiopia, while 72% of cases recovering on the 14th day of detection and 8% of cases requiring hospitalization. A clinical characterization study in China found out that 15.7% of COVID-19 cases develop severe disease and 1.4% of them died [11], which is slightly higher than our study. The case fatalities of Italy, China, and Spain during March 2020 were 4.77%, 3.77%, and 4.55% respectively [17]. This shows that the case fatality rate in Ethiopia is lower compared to other countries. This difference might be because the pandemic in Ethiopia is in the early phase. As compared to our

**Table 3. Secondary infection rate of COVID-19 among close contacts by contact setting in Ethiopia, 15 May to 15 June 2020.**

| Contact setting | All contacts | Positive contacts | Percent (%) | 95% LCL | 95% UCL |
|---|---|---|---|---|---|
| Household | 221 | 100 | 45.3% | 38.6% | 52.1% |
| Health care | 7 | 3 | 42.9% | 9.9% | 81.6% |
| Workplace | 42 | 13 | 30.9% | 17.6% | 47.1% |
| Others | 30 | 10 | 33.3% | 17.3% | 52.8% |
| Total | 300 | 126 | 42% | 36.4% | 47.8% |

Among ten sub-cities in Addis Ababa City, the highest number of COVID-19 cases, close contacts, and close contacts tested COVID-19 positive by RT-PCR were reported from Lideta Sub-city.

study the recovery period is longer in China, the mean recovery date is 24.7 days, as indicated in the estimation of COVID-19 severity. This study also estimated the age specific hospitalization rate ranges from 1.1% in the age group 20–29 years up to 18·4% in those 80 years and older [18]. A study on epidemiology and transmission of 391 cases and 1286 of their close contacts in Shenzhen, China showed 80% of COVID-19 cases were symptomatic and only 3% were severe and require hospitalization [19].

Of the 100 cases enrolled in our study, two of them (2%) required mechanical ventilation during their treatment. A study among the first 33 COVID-19 cases in Ethiopia also showed 12.1% of cases required mechanical ventilation [20] and another study in Germany indicated that the proportion of COVID-19 patients who required mechanical ventilation were higher (17%) compared to our finding [21]. This could be attributed to intensified community-based surveillance in Addis Ababa following the increasing report of COVID-19 cases in the city. This enables to detect cases timely and initiate treatment before worsening of the disease. Whereas, a study in Beijing, China showed that 2.7% of cases were critically ill [22], which in line with the findings in our study.

In this study, from a total of 300 close contacts, 126 (42%) of the contacts were tested positive for SARS-CoV-2 infection, with the lowest rate in those exposed at the workplace (31%) and highest among household contacts (45%). In our study, the secondary clinical attack rate was 11.7% and is consistent with the secondary clinical attack rate among household contacts reported from Guangzhou, China ranging from 12.4% to 17.1% [23]. By contrast, in Taiwan, the secondary clinical attack rate among close contacts was very low (0.84%) compared to the finding in our study [24]. Another study in China showed that among the close contacts of one case, 35% were infected with COVID-19 which is comparable with the secondary attack rate in this study [25]. In the United States of America, among 445 close contacts who were actively monitored during the initial phase of the pandemic in the country, the symptomatic secondary attack rate was 10.5% which is very close to the secondary clinical attack rate in our study [26]. These differences might be due to the difference in testing strategy, in Ethiopia laboratory test is being done for all close contacts regardless of symptom.

According to the findings of the present study, the incubation period ranges from 2 to 14 days and the median incubation period is found to be seven days and is consistent with a report by Sarda and colleague that the COVID-19 incubation period ranging from 1 to 14 days, with an average of 5 to 6 days. More than 97% of the COVID-19 cases experience symptoms within 14 days [27]. Similar findings have also been reported by Qin-Long Jing et.al, Stephen A. Lauer et.al and Zuopeng Xiao et.al, who reported the incubation period ranging from 2 to 14 days with a mean or median incubation period of 5 to 9 days [23, 28, 29].

The median time between the onsets of the symptom of the primary case and the corresponding secondary cases was 11 days (5 to 26 days), according to our study. The mean serial interval in Singapore and Tianjin of China was 5.2 and 3.9 days, which are shorter compared to this study finding [30]. A study by Hiroshi Nishiura et.al also calculated a serial interval, by synthesizing published studies and technical reports, that is shorter than our finding (median serial interval of 4.6 days) [31].

Our study finding showed that the basic reproduction number ($R_O$) in Ethiopia is 1.26, which indicated the average number of secondary cases for a primary case was 1.26. This finding indicates that the current transmission rate in the country is not much far from the level of control, as the basic reproduction number at the level of control was less than one [32]. In many other settings, the basic reproduction number is much higher than in the case of Ethiopia. The basic reproduction number in the Republic of Korea ranges from 2.6 to 3.2; in Italy, it ranges from 2.6 to 3.3, even in Africa it is 2.37 [33, 34]. This could be due to the early initiation of massive preventive and control measures in the country that include mandatory quarantine

of passengers, mandatory use of facemask in public places, closing of a non-essential business, banning of public gathering, transportation with half capacity, and early closure of schools. However, the basic reproduction number in this study was higher than China after February 11, 2020 which is 0.20 in Hubei 0.05 outside Hubei province [35].

## Limitations of the study

This study was conducted in the capital city of the country in which it might not represent the nationwide COVID 19 transmission dynamics. In addition, there might be rare conditions that close contacts may develop symptoms after 14[th] days of follow up.

## Conclusion

We determined major epidemiological and clinical outcomes of COVID-19 cases and close contacts in Ethiopia. The proportion of asymptomatic infection, as well as secondary infection rate among close contacts, were higher in this study compared to other studies. By contrast, the longer serial interval and lower basic reproduction number favor the slow progression of the pandemic. This all gives a wider window of opportunities to control the spread and to mitigate the negative impact of the pandemic on the already constrained health system. Hence, we would like to emphasize the need to intensify testing, prevention, and control measures through the involvement of all relevant stakeholders.

## Supporting information

**S1 File.**
(XLSX)

## Acknowledgments

The authors would like to thank an Ekka Kotebe and Millennium Hall COVID-19 treatment centers for facilitating data collection. We would like to thank all the study participants.

## Author Contributions

**Conceptualization:** Shambel Habebe Watare, Faiqa Ebrahim, Musa Emmanuel, Zewdu Assefa, Aschalew Abayneh, Ebba Abate.

**Data curation:** Mikias Alayu Alemu, Negussie Yohannes, Dagnachew Mulugeta.

**Formal analysis:** Shambel Habebe Watare, Mikias Alayu Alemu, Adamu Tayachew, Negussie Yohannes.

**Investigation:** Shambel Habebe Watare, Mikias Alayu Alemu, Adamu Tayachew, Lehageru Gizachew, Adisu Kebede, Asdesach Tessema, Wubayehu Kassa, Mengistu Biru, Mikias Teferi, Naod Wendrad, Kirubel Eshetu, Gizaw Teka, Habtamu Yimer, Abdulhafiz Hassen, Mesfin Wosen, Mukemil Hussen, Yaregal Fufa.

**Methodology:** Shambel Habebe Watare, Faiqa Ebrahim, Musa Emmanuel, Zewdu Assefa, Aschalew Abayneh, Ebba Abate.

**Software:** Negussie Yohannes, Dagnachew Mulugeta.

**Supervision:** Shambel Habebe Watare, Adisu Kebede, Asdesach Tessema, Wubayehu Kassa, Mengistu Biru, Mikias Teferi, Naod Wendrad, Kirubel Eshetu, Gizaw Teka, Habtamu Yimer, Musa Emmanuel, Getachew Tolera, Zewdu Assefa, Aschalew Abayneh, Ebba Abate.

**Validation:** Shambel Habebe Watare, Negussie Yohannes, Wubayehu Kassa, Mikias Teferi, Kirubel Eshetu, Faiqa Ebrahim, Dagnachew Mulugeta.

**Visualization:** Negussie Yohannes, Dagnachew Mulugeta.

**Writing – original draft:** Shambel Habebe Watare, Mikias Alayu Alemu, Negussie Yohannes, Adisu Kebede, Zewdu Assefa.

**Writing – review & editing:** Shambel Habebe Watare, Mikias Alayu Alemu, Adamu Tayachew, Lehageru Gizachew, Adisu Kebede, Asdesach Tessema, Wubayehu Kassa, Mengistu Biru, Mikias Teferi, Naod Wendrad, Kirubel Eshetu, Gizaw Teka, Habtamu Yimer, Faiqa Ebrahim, Alemnesh Mirkuze, Saro Abdela, Musa Emmanuel, Zewdu Assefa.

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
