## [Decision Letter · Decision Letter 0]

8 Jul 2022

PONE-D-22-10807An Investigation of a Hundred COVID-19 Cases and Close Contacts in Ethiopia, May to June, 2020: A Prospective Case-ascertained StudyPLOS ONE

Dear Dr. Watare,

Thank you for submitting your manuscript to PLOS ONE. After careful consideration, we feel that it has merit but does not fully meet PLOS ONE’s publication criteria as it currently stands. Therefore, we invite you to submit a revised version of the manuscript that addresses the points raised during the review process.

We look forward to receiving your revised manuscript.

Kind regards,

Sebsibe Tadesse, PhD

Academic Editor

PLOS ONE

 “This study was conducted as part of the routine national COVID-19 response efforts under the Ethiopian Public Health Institute. Thus, The author(s) received no specific funding for this work.”

5. We note that Figures 1, 2 and 3 in your submission contain map images which may be copyrighted. All PLOS content is published under the Creative Commons Attribution License (CC BY 4.0), which means that the manuscript, images, and Supporting Information files will be freely available online, and any third party is permitted to access, download, copy, distribute, and use these materials in any way, even commercially, with proper attribution. For these reasons, we cannot publish previously copyrighted maps or satellite images created using proprietary data, such as Google software (Google Maps, Street View, and Earth). For more information, see our copyright guidelines: http://journals.plos.org/plosone/s/licenses-and-copyright.

 a. You may seek permission from the original copyright holder of Figures 1, 2 and 3 to publish the content specifically under the CC BY 4.0 license. 

Reviewers' comments:

Reviewer's Responses to Questions

**Comments to the Author**

1. Is the manuscript technically sound, and do the data support the conclusions?

Reviewer #1: Partly

2. Has the statistical analysis been performed appropriately and rigorously? 

Reviewer #1: I Don't Know

3. Have the authors made all data underlying the findings in their manuscript fully available?

Reviewer #1: No

4. Is the manuscript presented in an intelligible fashion and written in standard English?

Reviewer #1: Yes

5. Review Comments to the Author

Reviewer #1: These are my comments for a manuscript entitled: An Investigation of a Hundred COVID-19 Cases and Close Contacts in Ethiopia, May to June, 2020: A Prospective Case-ascertained Study.

Overall, there is indeed value to publish this manuscript, as a way to promote research in the African region & also adding another data point for WHO's FFX protocol usage in Ethiopia. However, there are several critical issues that the authors should address, particularly in the methods section.

Critically, I find it worrying that there is no mention at all, on the definitions for the measures reported in this manuscript. Measures such as the secondary attack rate, secondary clinical attack rate, incubation period, serial interval (SI) should be included in the Statistical analysis sub-section. How was the reproduction number estimated? Also, how were the 10 clusters identified for SI calculation? Puzzlingly, the 10 clusters were only mentioned in the results.

Please also clarify the following for the methods sections:

1. How were the 3 contacts "randomly selected" for the study? If household size information was also collected, I suggest to report them in your results, so that international readers can roughly gauge the impact of choosing 3 contacts rather than enrolling all contacts into the study.

2. Please mention also the contact settings included in the study (this is only mentioned in table 5 results).

3. As it is mentioned that "a review of medical records" was done, clarify as well what was reviewed? Is this where co-morbidity conditions were checked for all cases?

4. It is mentioned that a symptom diary was collected. What was the use of this diary? If contacts develop symptoms during the 14 day follow-up, were they tested?

5. Please clarify if ethical review was done for this study. Or if it is waived as the study was conducted as part of the routine national COVID-19 response efforts under the Ethiopian Public Health Institute. If the latter is true, do include it in the ethics subsection of the main text.

For the results section, some tables are not informative to justify its use; it can be removed and written in text format. Such as Tables 2 & 3.

For the discussion section,

1. Check if the studies used to be compared with this study's findings are in fact comparable. Because it doesn't seem reasonable to compare proportions from this study (n=100) against much larger studies (n=1000 to 10000). I suggest to compare against existing FFX results from other countries.

2. Do mention any limitations for this study.

3. "This will facilitate timely estimates of the severity and transmissibility of COVID-19 infection. The finding will also inform public health responses and policy decisions in developing appropriate response strategies." I agree of the study's appropriateness to fulfil the above-mentioned study rationale. It will be interesting for international readers to learn what happen in Ethiopia (or at the capital city), after the first 100 cases. Were the results translatable to what happened after first 100? And if these result were ready in the timely manner, how did it help inform public policy makers at that time?

4. Can you suggest how would these results help to inform public policy makers under the current Omicron variant circulation?

6. PLOS authors have the option to publish the peer review history of their article (what does this mean?). If published, this will include your full peer review and any attached files.

Reviewer #1: No

---

## [Author Response · Author response to Decision Letter 0]

17 Sep 2022

We would like to acknowledge the reviewer for the critical review. We have included all the required information as per the reviewer’s comment/suggestion in the method part. 

The ten clusters are selected for SI calculation because only contacts in these clusters tested positive and transmission of disease from index to contacts is required to perform SI estimation/calculation.

The three contacts were selected randomly using lottery method, we have now included it in the method section. However, the household size information was not collected. It is also not indicated in WHO’s FFX protocol.

The contact settings incorporated under the study population and period sub section of the method section.

We clarify about which medical record has been reviewed in the data collection sub section. Yes, the comorbidities for all cases were checked in this record but also all the cases were asked for comorbidities during interview.

Symptom diary was done to check if the contacts develop symptom during the follow up period and will be tested if they develop any sign and symptoms of COVID 19.

We have clarified that the ethical review is waived as the study was conducted as part of the routine national COVID-19 response efforts under the Ethiopian Public Health Institute.

We compared our findings with similar publications of comparable study participants. In addition, we have also discussed our findings comparing with studies of similar subject matter but high number of study participants. Our study participants are also 400 (100 cases and 300 close contacts).

We do include the limitations for this study at the last paragraph of the discussion section. 

The result of this study was available on time and the national COVID 19 preparedness and response plan and the response strategy was revised based on this finding. 

Thanks for this very important question. Indeed, this study was done without considering the variants of COVID 19.

The detail explanation is attached in the Rebuttal letter in this submission.

---

## [Editor Report · Decision Letter 1]

20 Sep 2022

An Investigation of a Hundred COVID-19 Cases and Close Contacts in Ethiopia, May to June, 2020: A Prospective Case-ascertained Study

PONE-D-22-10807R1

Dear Dr. Shambel Habebe Watare,

We’re pleased to inform you that your manuscript has been judged scientifically suitable for publication and will be formally accepted for publication once it meets all outstanding technical requirements.

Kind regards,

Sebsibe Tadesse, PhD

Academic Editor

PLOS ONE

---

## [Editor Report · Acceptance letter]

6 Oct 2022

PONE-D-22-10807R1 

An investigation of a hundred COVID-19 cases and close contacts in Ethiopia, May to June, 2020: A prospective case-ascertained study 

Dear Dr. Watare:

I'm pleased to inform you that your manuscript has been deemed suitable for publication in PLOS ONE. Congratulations! Your manuscript is now with our production department. 

Kind regards, 

on behalf of

Dr. Sebsibe Tadesse 

Academic Editor

PLOS ONE